# Climate, inter-serotype competition and arboviral interactions shape dengue dynamics in Thailand
Lester J. Perez [1,2] ✉, Julie Yamaguchi[1,2], Sonja Weiss [1,2], Christiane Carlos [1,2], Todd V. Meyer[1,2], Mary A. Rodgers[1,2], Pakpoom Phoompoung[3], Yupin Suputtamongkol[3], Gavin A. Cloherty[1,2] & Michael G. Berg [1,2]

The incidence and global spread of dengue are reaching alarming levels. Thailand represents a critical disease epicenter and demands an understanding of the environmental and evolutionary pressures that sustain DENV transmission. Unlike most affected countries experiencing recurrent outbreaks of the same serotype or replacement of one serotype for another, Thailand is an ecological niche for all four serotypes. Favorable climate and mosquito vector availability maintain a landscape defined by stable, endemic circulation of genotypes, with minimal genetic variation attributed to sporadic, external introductions. This equilibrium is achieved through inter-serotype competition, characterized by reproductive fitness levels that maintain infections (Re>1) and elevated evolutionary rates ($\sim 10^{-4}$), which steadily increase the genetic diversity of each serotype. This conclusion is reinforced by the identification of numerous positively selected mutations, skewed in the direction of non-structural proteins conferring replication and transmission advantages versus those present in structural proteins evading neutralizing antibodies. Precipitous drops in DENV cases following outbreaks of Chikungunya suggest that interactions with other arboviruses also impact DENV dynamics through vector competition, replication inhibition or partial cross-protection. Thailand is a major exporter of DENV cases and novel emergent lineages gaining fitness here are likely to spread internationally. Surveillance is therefore paramount to monitor diversification trends and take measures to avoid the establishment of similar sustained, local transmission in other countries.

Dengue virus (DENV), a member of the *Flaviviridae* family and genus Flavivirus[1], continues to pose a severe global health threat[2] with an estimated ~400 million infections occurring annually[3] across more than 130 countries[4]. As a species, DENV includes four distinct serotypes (DENV-1, DENV-2, DENV-3, and DENV-4), each capable of causing a range of symptoms from mild fever to severe and potentially fatal hemorrhagic fever[5]. Dengue's impact is notably significant in tropical and subtropical regions where ecological conditions favor the proliferation of its primary vector, the *Aedes* mosquitoes[6]. In recent decades, the geographic distribution of dengue has expanded[7], influenced by factors such as urbanization[8], global travel[9,10], and climate change[11], leading to widespread epidemics and increased healthcare burdens[12]. The World Health Organization (WHO) has classified dengue as a major international public health concern, reflecting its escalating incidence[4] and the challenges it presents in terms of

effective vaccine development[13,14] and vector control strategies[15]. This ongoing scenario underscores the urgent need for continued research into the epidemiology, molecular biology, and control measures for dengue to mitigate its global impact.

In tropical and subtropical regions around the world, and especially in Thailand, the complexity of dengue epidemiology is further intensified by the phenomenon of antibody-dependent enhancement, which can exacerbate disease severity during secondary infections with a different serotype[16]. This dynamic interplay not only influences infection outcomes but also shapes the landscape of serotype co-circulation within the host population[17]. Moreover, Thailand's favorable ecological and climatic conditions provide a fertile ground for the proliferation of various arboviruses, such as Chikungunya, Zika, and Yellow Fever viruses, which are transmitted by the same *Aedes* species mosquitoes[18]. These arboviruses often co-circulate in the

[1]Infectious Disease Research, Abbott Diagnostics Division, Abbott Laboratories, Abbott Park, Lake Bluff, IL, USA. [2]Abbott Pandemic Defense Coalition (APDC), Abbott Park, Lake Bluff, IL, USA. [3]Faculty of Medicine, Siriraj Hospital Mahidol University, Bangkok, Thailand. ✉ e-mail: lester.perez@abbott.com

same regions, leading to intricate patterns of inter-viral competition and interaction. Understanding the competitive dynamics between different dengue serotypes, as well as their interactions with other circulating arboviruses, is crucial for unraveling the complexities of viral transmission and developing effective control strategies.

The Abbott Pandemic Defense Coalition (ADPC) has been surveilling causes of acute febrile illness (AFI) around the globe committed to the early detection and mitigation of infectious disease threats of pandemic potential[19]. Thailand represents a unique setting where DENV outbreaks are commonplace, and all four serotypes co-circulate. Here, we explored the intra- and inter-serotype dynamics of dengue virus and its interactions with other arboviruses in Thailand. By employing phylodynamic, statistical and predictive models, we sought to identify factors driving viral trends and their implications for disease spread and public health intervention strategies.

## Results

### Epidemiological Analysis of dengue virus infections relative to the climate-driven index

Before exploring the factors that determine dengue dynamics in Thailand, we first describe the magnitude of its burden and its typical yearly pattern. An analysis of dengue fever (DF) epidemiological trends in Thailand from 2014 to 2023 demonstrates a clear seasonality in case numbers, with yearly peaks coinciding with the rainy season and the periodicity of Index P, a climate-driven variable that reflects mosquito availability (Fig. 1A, B). Particularly high case counts were noted in 2015 and 2019, along with a significant decline in 2021. However, by 2023, the case numbers had surged to a maximum (Fig. 1B). Over the same 10-year period, deaths attributed to dengue fever were consistently low (0.1%). Indeed, mortality trends paralleled incidence rates, indicating that fluctuations in number of deaths were more indicative of changes in virus transmission (e.g. proportional to case counts) rather than alterations in clinical severity (Fig. 1B).

*IndexP*, a novel suitability index based on a climate-driven mathematical expression for the basic reproductive number of mosquitoes-borne viruses[20]. Index P has been previously used to estimate mosquito population dynamics and virus–mosquito transmission efficiency for dengue[20], yellow fever virus[21] and zika virus[22] in Brazil, as well as west nile virus[23] in Israel. In the current study Index P was calculated from several factors, including 1) climate: temperature, precipitation, humidity, 2) mosquito prevalence: rate of bites, mating, number of females, and 3) human susceptibility: number of bites * life expectancy. Thailand exhibits a distinct *IndexP* pattern, characterized by a gradual migratory shift from southern regions toward the north and back to the south, corresponding to peaks in March and September, respectively. During the peak period spanning July-October, summer conditions are most conducive to mosquito breeding and survival and create heightened potential for dengue transmission. As the year progresses, index values recede in the north to signal a decrease in abundance of mosquito populations with the higher values in the southern regions for the months of November through May. This cyclical movement of mosquito populations[24] suggests a strong seasonal component driven by climatic factors that undoubtedly drives dengue transmission (Fig. 1C and Supplementary Video 1).

While this spatiotemporal analysis seeks to approximate the relationship between ecological transitions and mosquito prevalence and has the potential to identify hotspots and periods of intensified dengue outbreak risk, it does not singularly account for the observed reduction in dengue cases in 2021 across Thailand. Additional factors may have contributed to this outcome: herd immunity, vector control initiatives, public health interventions, or changes in human behavior (e.g. reduced contact between human and mosquito due to covid lockdowns). To determine whether any of these anthropogenic factors influence infection trends in Thailand, we conducted a detailed, yearly analysis of the geospatial distribution of dengue cases from 2014 to 2023 and related this to population density (Fig. 1D–E and Supplementary. Video 2). During the 2014-2020 period, patterns mirrored that of the typical *IndexP* cycle, migrating from the south to more densely populated central and northern provinces where the most

significant aggregation of dengue cases were found. Notably, in 2021, the distribution of dengue cases was largely confined to the western regions with fewer inhabitants, and notably absent from the most populated urban areas. This unexpected containment pattern suggests that partial immunity may have offered protection against widespread transmission common to the central and northern provinces. By 2023 the pattern of distribution of cases have been restored (Fig. 1E).

### Dynamics and temporal diversification of dengue serotypes in Thailand

Understanding levels of genetic diversity and estimating trajectories will indicate which serotypes or lineages therein are expanding whereas others could be experiencing an evolutionary bottleneck that could explain the fluctuations in the number of DENV cases in the country. To investigate the genetic diversity of DENV in Thailand, we conducted a Maximum Likelihood (ML) phylogenetic analysis utilizing all available, nearly complete genomic sequences ( > 70% coverage) from across the globe present in public databases. We enriched this dataset by adding approximately 200 full genome sequences from Thailand obtained in this study by metagenomic NGS and viral target enrichment (CVRP), which includes representatives from all four serotypes. This comprehensive analysis aimed to delineate the evolutionary relationships and track the potential emergence of new viral variants over time in the country. Our phylogenetic analysis confirmed the endemic circulation of all four DENV serotypes within Thailand (Fig. 2A). Contrary to the findings suggested by Poltep et al.[25], which used partial sequences from envelope compared to our full genome analysis, we did not detect a high level of intra-serotype diversity among Thai strains. Rather, we observed the formation of homogenous clusters within the serotypes, which were composed of sequences from both previous studies (blue) and those characterized in our current research (maroon) submitted for this study. This phylogenetic pattern strongly suggests that the DENV landscape in Thailand is defined by stable, endemic circulation with sporadic introductions rather than by widespread genetic variation. The clusters formed by previous Thai sequences and reinforced by the addition of our current data containing strains from the same lineages, suggest a relatively constrained evolutionary dynamic within the country (Fig. 2A).

Our longitudinal analysis of the annual proportion of dengue serotypes (Fig. 2B), revealed a highly dynamic pattern in serotype prevalence over a 45-year span. The co-circulation of serotypes appears to be an established pattern in the country and raises the specter of DENV co-infections, which could influence disease manifestation and epidemic behavior. The early 1980s were marked by the predominance of DENV1, nearly vanishing in subsequent years, and then resurging and persisting at ~25% from the mid-1990s onwards. From 2016 onward, it has regained its status as the most prevalent serotype. DENV2 also exhibited epochs of dominance (mid-1980s to early 1990s) and absence (1991 to 1993), but it too maintained its presence at steady levels for 25 years (1995–2020). However, no genomic data for DENV2 was identified in 2021which should indicate a reduced circulation for this serotype since it reappears in 2022. DENV3 circulation was sporadic throughout the 80's, with several years of absence and other years where it appears to have been the only serotype present. Over the period of 1994-2018, yearly prevalence steadied at 10–25%, after which DENV3 seems to have largely disappeared (2021–2022) but reemerging in 2023. DENV4 showed a similar pattern as DENV3, characterized by occasional peaks during the 80's, consistent levels (5–50%) during the 1992-2018 period, and sporadic levels in recent years (Fig. 2B).

Time-calibrated phylogenetic trees further illustrate the concurrent circulation and evolutionary history of all four DENV serotypes in Thailand (Fig. 2C–F). Thai sequences, denoted by red dots on the trees, formed distinct clusters that signify periods of localized evolution, suggesting a scenario where endemic diversification has occurred. However, the distribution of these sequences could either indicate there were multiple introductions of DENV into Thailand or represent their persistent endemic presence punctuated by phases of expansion. The formation of discrete subclades within principal clades indicates there are lineage-specific attributes

that potentially impact transmission and pathogenicity (Fig. 2C–F). In the case of DENV1, there are 5 genotypes globally, but only two are present in Thailand. Most sequences belong to Genotype I (GI), with a smaller representation from Genotype II (GII) (Fig. 2C). In GI some clades persisted for decades whereas others went extinct after a short period of time. Genotype II (GII) sequences appear to have been introduced between 1961 to 1984 but failed to diversify within Thailand, as evidenced by the absence of emerging lineages from these sequences (Fig. 2C). A notable feature within the DENV1 GI strains is the ancestral composition of Thai sequences within each cluster, supporting the hypothesis that diversification of these clades

occurred primarily within Thailand and lead to their outward spread to other regions. To test this hypothesis, we constructed a temporal phylogeny to scrutinize the emergence and demographic expansion of this genotype in the country dating its emergence back to 1968 (Fig. 2C, *inset*). For DENV2, sequences were almost exclusively grouped into the Asian I (G-AI) genotype, with a smaller subset corresponding to introductions from the Asian-American (G-AA) and Cosmopolitan (G-C) genotypes, which did not show further diversification (Fig. 2D). Like DENV1-GI strains, the DENV2 G-AI strains constitute a monophyletic clade that emerged ~1966 (Fig. 2D, *Inset*). By contrast, DENV3's genetic landscape in Thailand was characterized by

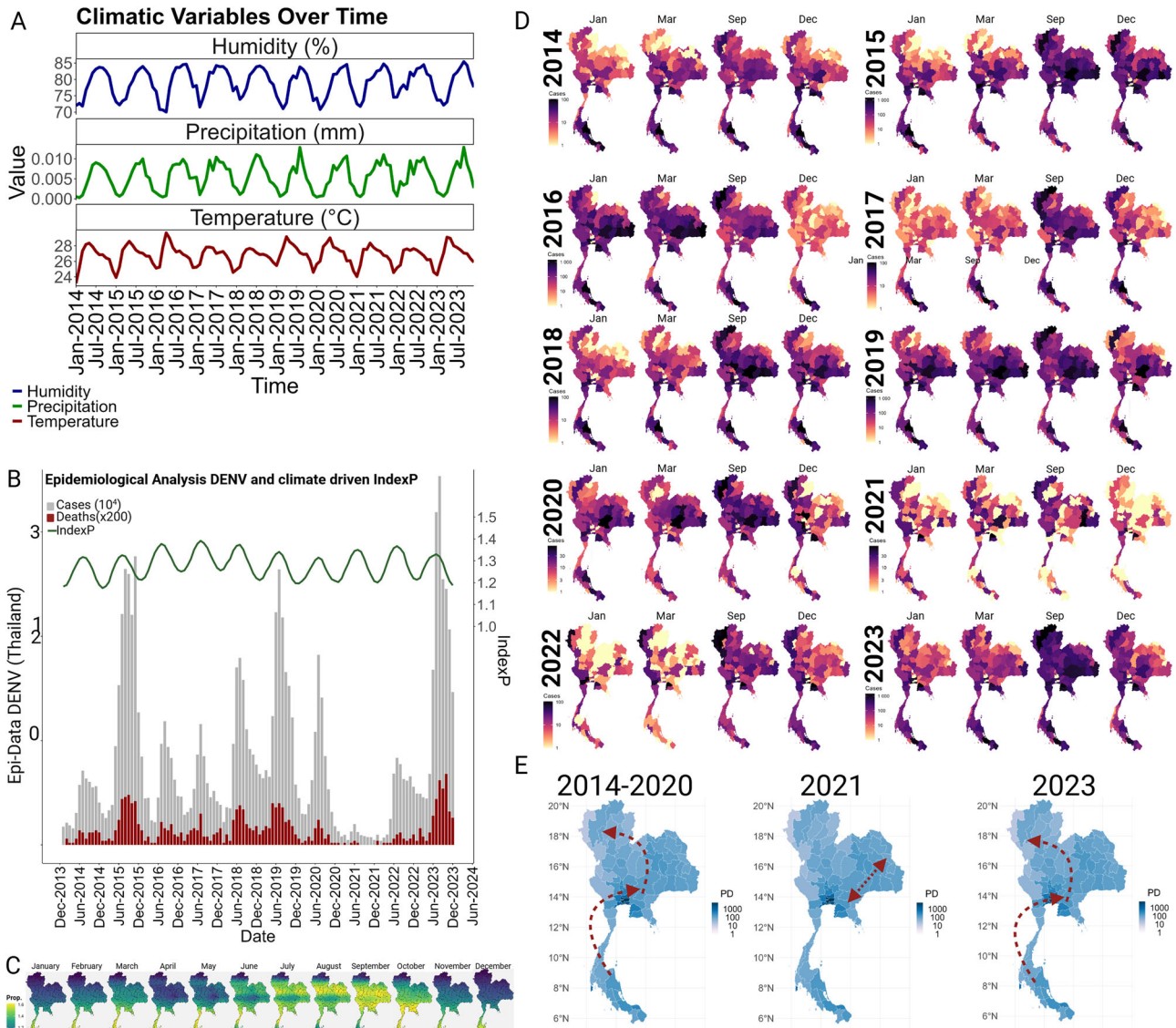

**Fig. 1 | Dynamics of dengue virus cases and mosquito-viral suitability Index (Index P), in Thailand 2014-2023. A** Climatic variables (precipitation, humidity and temperature) per month obtained for Thailand from 2014 to 2023 were used as priors to estimate index. **B** Temporal trends and climatic influences on dengue virus transmission represented by monthly reported cases and fatalities associated with DENV in Thailand, alongside with a mosquito-viral suitability index (Index P). Index P integrates climatic data, anthropogenic factors, and the prevalence of the primary vector, Aedes aegypti. This index is computed monthly and averaged nationally, reflecting the combined impact of environmental and human variables on vector capacity and virus transmission (for detailed methodology see Materials and Methods and Supplementary Fig. S1). **C** Spatiotemporal patterns of mosquito-viral suitability across Thailand showed by the average monthly values of Index P from 2014 to 2023 elucidating the geographical variation in vector suitability,

highlighting regions and times of heightened risk. For detailed monthly/year maps see animations provided in Supplementary Video 1. **D** Spatiotemporal distribution of DENV cases. Delineation of temporal dynamics of DENV transmission, identifying the geographic distribution of the cases per season considering the onset of the transmission in January, an initial peak in March, the highest peak in September, and the season's end in December (determined in panel A). The monthly mapped cases are available in Supplementary Video 2. **E** Synthesis of case trajectories place alongside with population density, offering insights into the spatial clustering of outbreaks, red arrows illustrate the directional progression of the cases over the specified period. The directional progression of cases were derived from statistically determined central positions (centroids) of case distributions using kernel density estimation (KDE) across successive time periods.

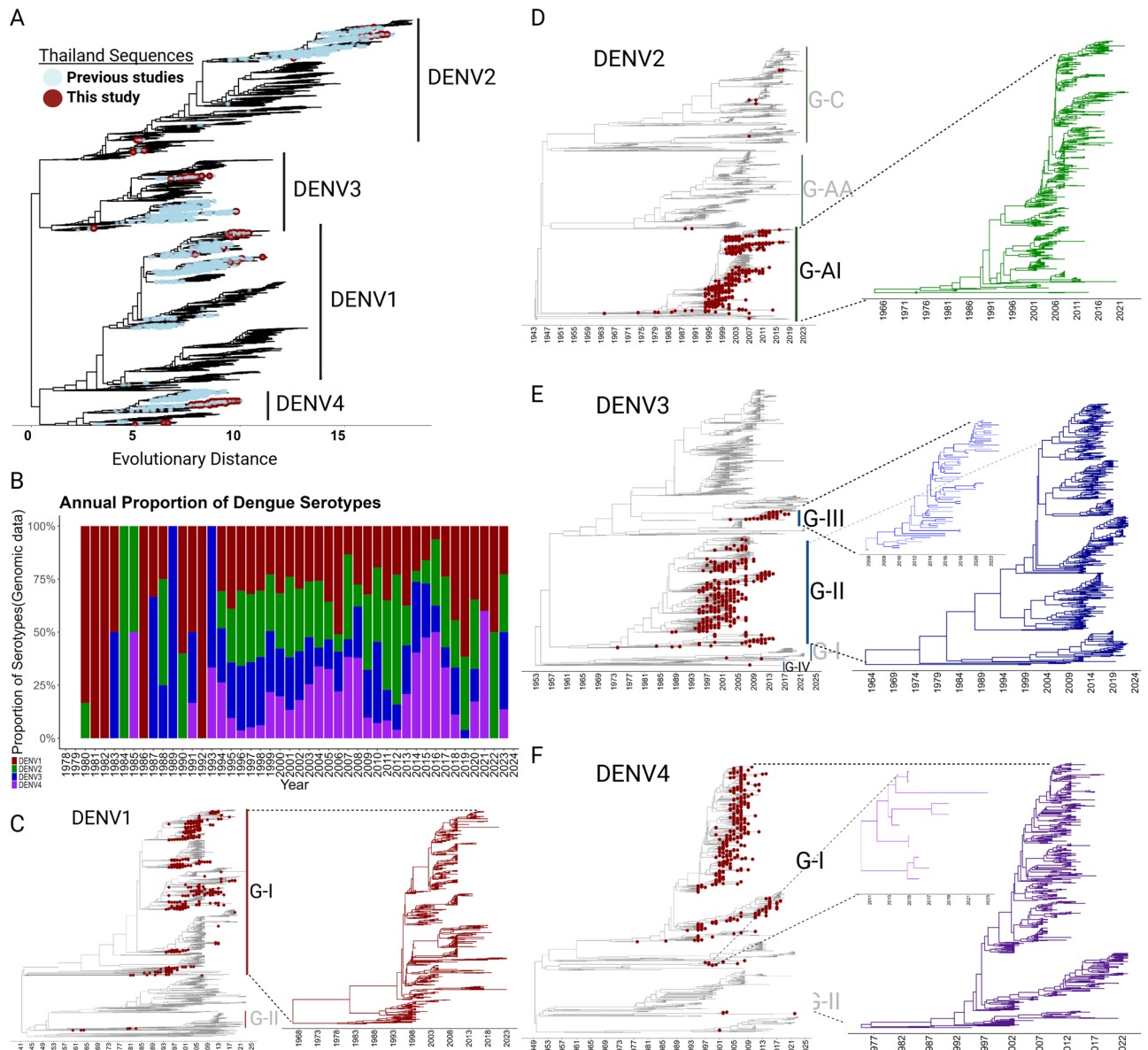

**Fig. 2 | Phylodynamics of dengue virus (DENV) in Thailand. A** Phylogenetic reconstruction of dengue Virus. Maximum likelihood (ML) phylogenetic tree, constructed from all sequences in Dataset A (see Materials and Methods). The tree illustrates the presence of all four DENV serotypes (DENV1-4) in Thailand, revealing patterns of diversification and cluster formation that suggest endemic circulation. Each serotype is denoted in the tree, and sequences from both previous studies and the current study are highlighted. **B** Serotype distribution and prevalence over time. This panel displays the distribution and frequency of the four dengue serotypes in Thailand from 1980 to 2023. It highlights the temporal shifts in serotype dominance and prevalence, reflecting evolving transmission dynamics, vector-host interactions, and population immunity levels. **C–F** Time-stamped phylogenies represented by MCC-tree for all four DENV serotypes. Sequences from Thailand are highlighting with red tips. Genotypes within each serotype that include sequences from Thailand are denoted (the distribution of all genotypes within each serotype please see Supplementary Material Fig. S1). Insets show detailed views of the endemic lineages in Thailand across the years to highlight the different evolutionary trajectories. Lineages that didn't share the same Thai ancestor were treated as distinct, and sequences without further diversification were considered external introductions (confirmed in Fig. 4).

the diversification of two primary lineages. Genotype II sequences have been established since 1977, whereas a secondary, smaller lineage of Genotype III sequences was introduced more recently in 2006, alongside Genotype IV strains that displayed limited diversification (Fig. 2E). DENV4 temporal phylogeny mirrors that of DENV1 and DENV2, with a predominant monophyletic cluster of Genotype I sequences circulating since 1977. A secondary, genetically distinct cluster within the same genotype emerged or was introduced in 2011. Additionally, sporadic sequences from Genotype II (GII) appear to have been introduced between 2001 and 2021, yet these did not undergo further diversification within Thailand, as suggested by the static lineage structure (Fig. 2F). It is unclear if these genotypes were

introduced to or emerged from Thailand, but as a result, all four serotypes have been present in the country since the late 60's.

The Bayesian birth-death skyline plot explicitly estimates the rate of transmission, recovery, and sampling allowing the inference of the effective reproductive number (Re) directly from genetic data[26]. This approach has been extensively used to described fitness competition between lineages in different viral species including SARS-CoV2 in Senegal[27], HIV in North America[28] and dengue in Brazil[29]. An estimation of the Re for all six endemic Thai lineages exhibited a Re > 1, accounting for the mean value, (Supplementary Material Fig. S2), indicating that on average, each infected individual is capable of transmitting the disease to more than one other

individual. These levels highlight the potential for not only sustained growth, but epidemic growth as well. Notably, the Re values across different lineages remained relatively constant over time, suggesting no discernible differential impact on the transmission dynamics among the lineages within Thailand. This consistency implies that the observed genetic diversity is likely the result of competitive interactions between lineages, rather than attributed to increased fitness or a transmission advantage. In agreement with the notion of an equilibrium having been reached between vectors and hosts is that we observe stable, elevated evolutionary rates for all six lineages (Supplementary Table S1). If one or more lineages had any evolutionary advantage or disadvantage over each other, these values for Re and mutational rates would cease to maintain uniformity. These observations are important for understanding the evolutionary pressures at work and suggests that inter-lineage competition may be a significant factor shaping the patterns of genetic diversity in the DENV lineages circulating in Thailand.

## Impact of pervasive and episodic selection on the genetic makeup of dengue serotypes in Thailand

To assess which mutations may have enabled each lineage to compete more effectively, a positive selection analysis was performed. Pervasive evolutionary changes, or those driven by forces acting continuously over time and across populations, were statistically significant at only one position (1286 in NS2A) in DENV1 (Fig. 3A and Supplementary. Data1). This amino acid, as indicated by the FuBAR analysis, has undergone multiple replacements (L1286F, V1286F, or Y1286F) in DENV1 strains that have circulated in Thailand since 1992 (Fig. 3A, central panel). By contrast, an aBSREL analysis for measuring episodic selection, driven by sporadic or distinct events, identified 23 branches that were significantly affected ($p \leq 0.05$) (Fig. 3A, central panel, Supplementary. Data2). Subsequent MEME analysis identified 45 sites under episodic selection, with a noticeable preference for nonstructural proteins. Indeed, 13 substitutions in non-structural proteins were maintained over time compared to 4 in structural proteins (Fig. 3A, right panel, Supplementary. Data3 and Supplementary. Data13). For DENV2, the Fubar analysis indicated pervasive selection once again at only one position (2762 in the RdRp; NS5) (Fig. 3B and Supplementary. Data4 and Supplementary. Data13), with replacements V2762I, T2762I, K2762I, or A2762I observed in post-1991 Thai strains (Fig. 3B, central panel). The aBSREL analysis revealed 10 branches under episodic selection ($p \leq 0.05$) (Fig. 3B, central panel, Supplementary. Data5) and MEME pinpointed 15 sites under episodic selection, again predominantly in nonstructural proteins, specifically NS1 and NS5 (Fig. 3B, right panel, Supplementary. Data6 and Supplementary. Data13). For DENV3, pervasive selection was evident at position 412 in the envelope protein E (Fig. 3C and Supplementary. Data7 and Supplementary. Data13), leading to the H412Y replacement. This mutation was unique among serotypes for its impact on a structural protein and also unlike other serotypes, both pervasive and episodic selections were restricted to specific sub-lineages (Fig. 3C, central panel, Supplementary. Data8 and Supplementary. Data13). MEME analysis confirmed several sites were under episodic selection, predominantly in nonstructural proteins, except for the positively selected site K104R in the capsid (Fig. 3C, right panel, and Supplementary. Data9 Supplementary. Data13). DENV4 strains exhibited pervasive selection at three positions: 2255 in NS4A, and 2688 and 3118 in NS5 (Fig. 3D and Supplementary. Data10 and Supplementary. Data13). These sites, particularly in RdRp, were widespread across the phylogenetic tree, suggesting an effective replicative advantage. Interestingly, the NS4A site was limited to strains circulating between 2004 and 2014, reflecting a transient advantage (Fig. 3D, right panel). Only two branches, coinciding with a significant CHIKV outbreak in 2008-2009[30], were identified under episodic selection (denoted by orange dashed lines) (Fig. 3D, Supplementary. Data10). Similar to the other serotypes, MEME analysis confirmed several sites in nonstructural proteins were under episodic selection (Fig. 3D, right panel, and Supplementary. Data12 and Supplementary. Data13). Despite decades of circulation, only a few mutations were positively selected for each serotype. These resided primarily in non-structural proteins, suggesting that inter-serotype

competition in Thailand is driven by replicative advantages rather than immune evasion.

## Phylogeographic origins and dispersal patterns of dengue serotypes in Asia demonstrates that endemic circulation has driven diversification in Thailand

The monophyletic clustering of lineages and the presence of Thai strains at ancestral nodes suggest that these lineages may have originated in Thailand. (Fig. 2). To further substantiate the premise that competition between resident lineages has sustained DENV circulation in Thailand and that external introductions have played a minimal role, a discrete phylogeographic analysis was undertaken. Maximum Clade Credibility (MCC) trees (Fig. 4A) were constructed to visualize the transmission patterns and geographical dispersal of the four DENV serotypes beginning with their emergence to the present day. According to the phylogenetic rooting, the initial emergence of all four serotypes is traced back to Asia, with DENV1 and DENV3 originating from the Philippines, DENV2 from Papua New Guinea, and DENV4 from India, spanning the period from 1944 to 1955. The MCC trees demonstrate temporal clusters of geographically diverse sequences, indicating periods of increased transmission and spread. These clusters reflect not only persistent endemic circulation within certain regions but also cross-border dissemination of the virus. Distribution and phylogenetic branching patterns are known to align with human movement, vector ecology, and environmental conditions conducive to DENV propagation[9,10,31]. The analysis substantiates our initial hypothesis that the majority of the DENV diversification in Thailand can be attributed to sustained local transmission. This is determined by the identification of internal nodes which designate Thailand as the location of diversification for several main clades. Conversely, Thai sequences that cluster within genotypes less represented in the region, and which exhibit limited diversification, are indicative of external introductions into the country (Fig. 4A).

The Markov jump count analysis, as shown in the circular plots of Fig. 4B, quantifies and visualizes the transmission events (not total number of sequences) of each dengue virus (DENV) serotype between countries. Here, Singapore emerges as a significant focal point for the global dissemination of DENV, implicated in numerous exportation events involving all four serotypes. The circular plots suggest that Thailand is a substantial contributor to the international spread of DENV1, DENV2, and DENV3. However, the transmission of DENV4 from Thailand appears to be predominantly regional, with most dissemination events involving only neighboring countries such as Myanmar, Cambodia, and Singapore. Corroborating the insights obtained from the MCC trees, the Markov jump analyses further support the observation that Thailand has a relatively low incidence of DENV importation from external sources. Not surprisingly, the primary contributors are its neighboring countries, Myanmar, Cambodia, and Singapore. This pattern underscores the regional dynamics of DENV transmission and highlights the interconnected nature of DENV spread within Southeast Asia.

## Impact of CHIKV on genetic diversity and Force of infection on dengue Serotypes in Thailand

Finally, we asked what factors or events could account for the dearth of DENV infections in 2021 (Fig. 1A). Considering the potential impact of the CHIKV outbreak of 2009-2010 on the selection pressure of DENV4 (Fig. 3D) we first evaluated the trends in genetic diversity over time for all four serotypes in Thailand. Bayesian Skygrid reconstructions (Fig. 5A) represent the effective population sizes (Neτ) across DENV serotypes and lineages over an extended time frame (1967-2023) which provide insights into the genetic diversity and transmission dynamics of DENV within Thailand. DENV1 began its upward surge in 1994, with DENV2-4 serotypes following suit a few years later, each indicative of increased viral transmission and expansion in the country. What is notable in these Neτ plots are the 'pauses' or 'plateaus' that have interrupted the upward trajectories or sharp increases that have accelerated them, influenced by cyclical variations in transmission, environmental changes, intervention strategies,

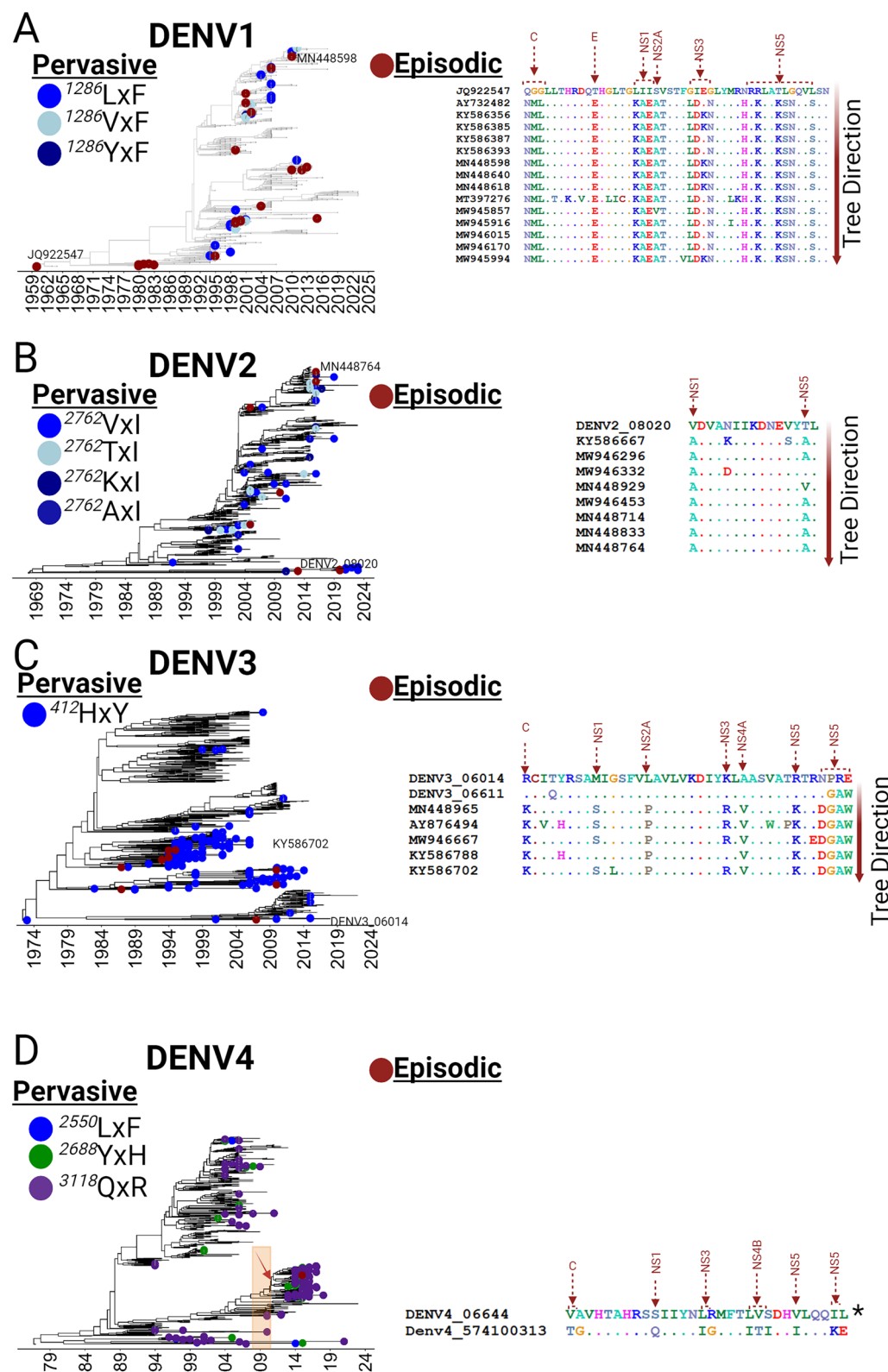

or inter-serotype competitions, which affect the equilibrium between rates of new infections and clearance. Upward jumps in diversity are required for a lineage to survive after events that quelled infections. DENV1 and DENV4 (lineage 1) exhibit inflection points around 2009 (dashed line; Fig. 5A), wherein these drastic changes in genetic diversity coincided with the major outbreaks of CHIKV reported in Thailand[30,32]. A similar flattening of

the genetic diversity curve between 2018-2020 was observed for all four serotypes (Fig. 5A).

Recent studies by Brito et al.[33] and Pinotti et al.[34] have suggested that short-term immunity induced by Zika virus (ZIKV) infections may contribute to the temporary suppression of dengue virus (DENV) cases. Our analysis aligns with these findings, showing an increase in ZIKV cases in

**Fig. 3 | Dynamics of positive selection across dengue virus serotypes circulating in Thailand.** Selection dynamics within the coding regions of four dengue virus (DENV) serotypes (**A**) DENV1, **B** DENV2, **C** DENV3, and **D** DENV4 as obtained from the time-stamped phylogenies. Each panel displays a Maximum Clade Credibility (MCC) tree at the center, trees were obtained from the coding regions of each DENV serotype (see Materials and Methods for details). The left panel highlights pervasive positively selected sites for each serotype, identified using the FUBAR method (see Supplementary Data 1–13). Each positively selected site is represented by a unique color, with varying shades indicating different amino acid replacements. These sites are integrated into the time-stamped phylogenies to trace the timing of each selection event accurately. The right panel shows episodically selected sites determined by the MEME method, with sites mapped onto specific branches that were identified using the aBSREL method for detecting branch-specific selection (see Supplementary Data 1–13). Sites that persisted throughout the evolutionary history are marked with arrows. Additionally, the locations of the viral proteins are also indicated on the trees (see Supplementary Data 14, mapping). Strains within branches identified under episodic positive selection are represented with the red dots across all the temporal trees (see Supplementary Data 1–13). (*Branches belonged to the same node therefore there is not a clear evolutionary direction).

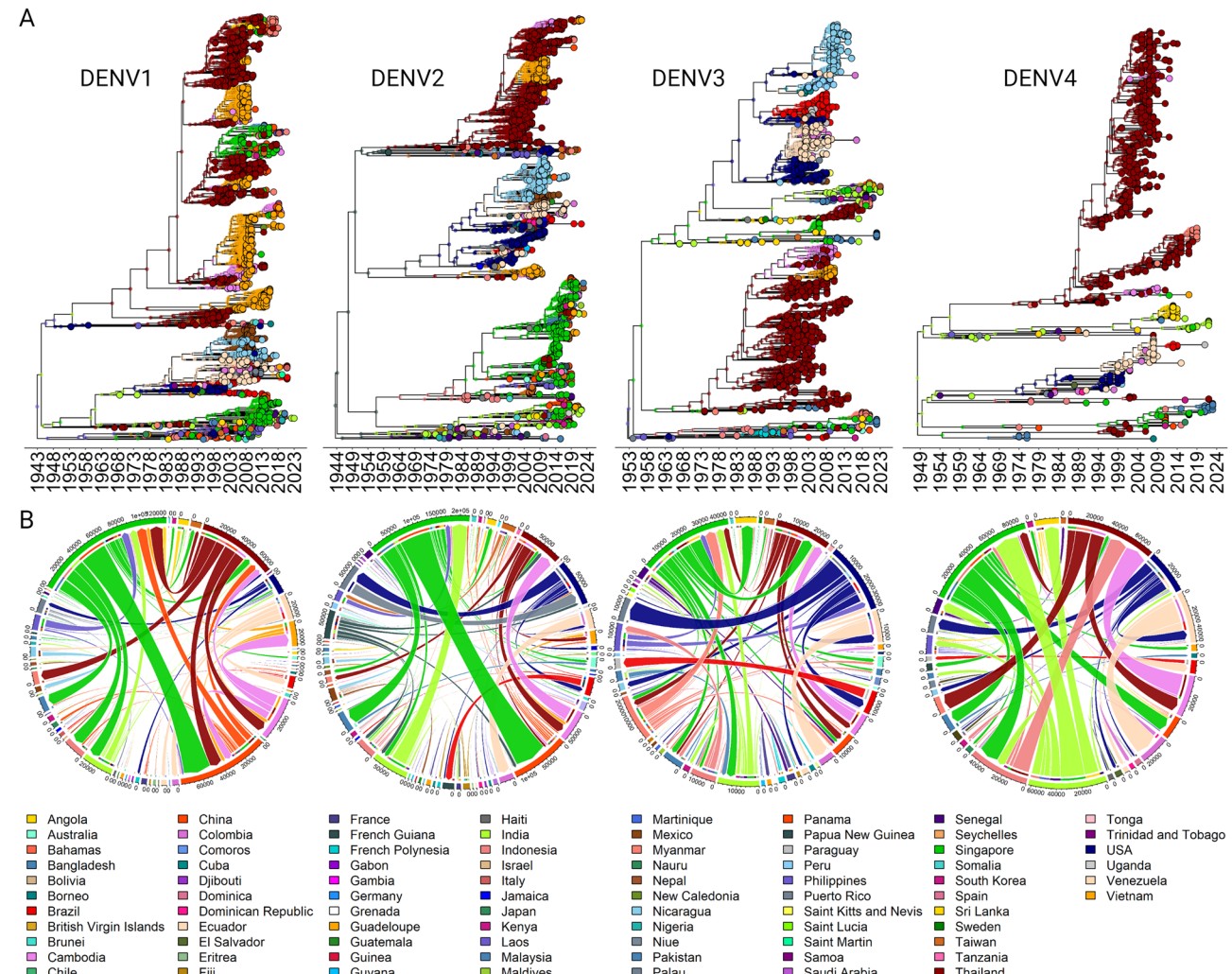

**Fig. 4 | Discrete phylogeographic analysis and Markov-jump trajectories of dengue virus (DENV) serotypes. A** Phylogeographic Relationships displayed using Maximum Clade Credibility (MCC) trees, this analysis elucidates the endemic diversification of DENV in Thailand and reveal the trajectories of all four DENV serotypes across the globe. The time-scaled phylogenies display ancestral nodes and current geographic locations (tips) as discrete states, illustrating the spatial and temporal spread of the virus. **B** Dynamic Pathways of Geographical Movement displayed as circular plots generated by the *'circlize'* package in R to depict the Markov-jump trajectories of DENV movement. The visualization highlights the frequency and routes of viral importation, exportation, and intra-country dispersal, providing the role of Thailand the global dispersal of DENV. (Color legend for all the countries is provided at the bottom of the Figure).

2020 (Fig. 5B) preceding the significant decline in DENV cases in 2021 (Fig. 5B). However, given the relatively low overall incidence of ZIKV across Thailand, it seems unlikely that ZIKV alone could confer widespread partial protection against dengue. Interestingly, Brito et al.[33] also reported a surge in chikungunya virus (CHIKV) cases prior to the decrease in DENV cases. Despite CHIKV (*Togavirus*) being taxonomically more distant from DENV than Zika virus (both flaviviruses), the temporal co-circulation is noteworthy. There were three consecutive outbreaks of CHIKV infections from 2018 to 2020, peaking in 2020, just before the significant reduction in DENV cases observed in 2021 (Fig. 5B). Other illnesses monitored at this time did not experience a similar drop in cases including scrub typhus[35] and leptospirosis[36], suggesting reporting of DENV would not have been deficient during the SARS-CoV-2 pandemic. Whether other factors related to mobility, such as travel restrictions, sheltering in place orders, and limited gatherings played a role cannot be ruled out[37]. Nevertheless, the sequential pattern of viral epidemics hints at complex interactions between circulating arboviruses that could drive down cases through cross-immunity or competition for vector resources.

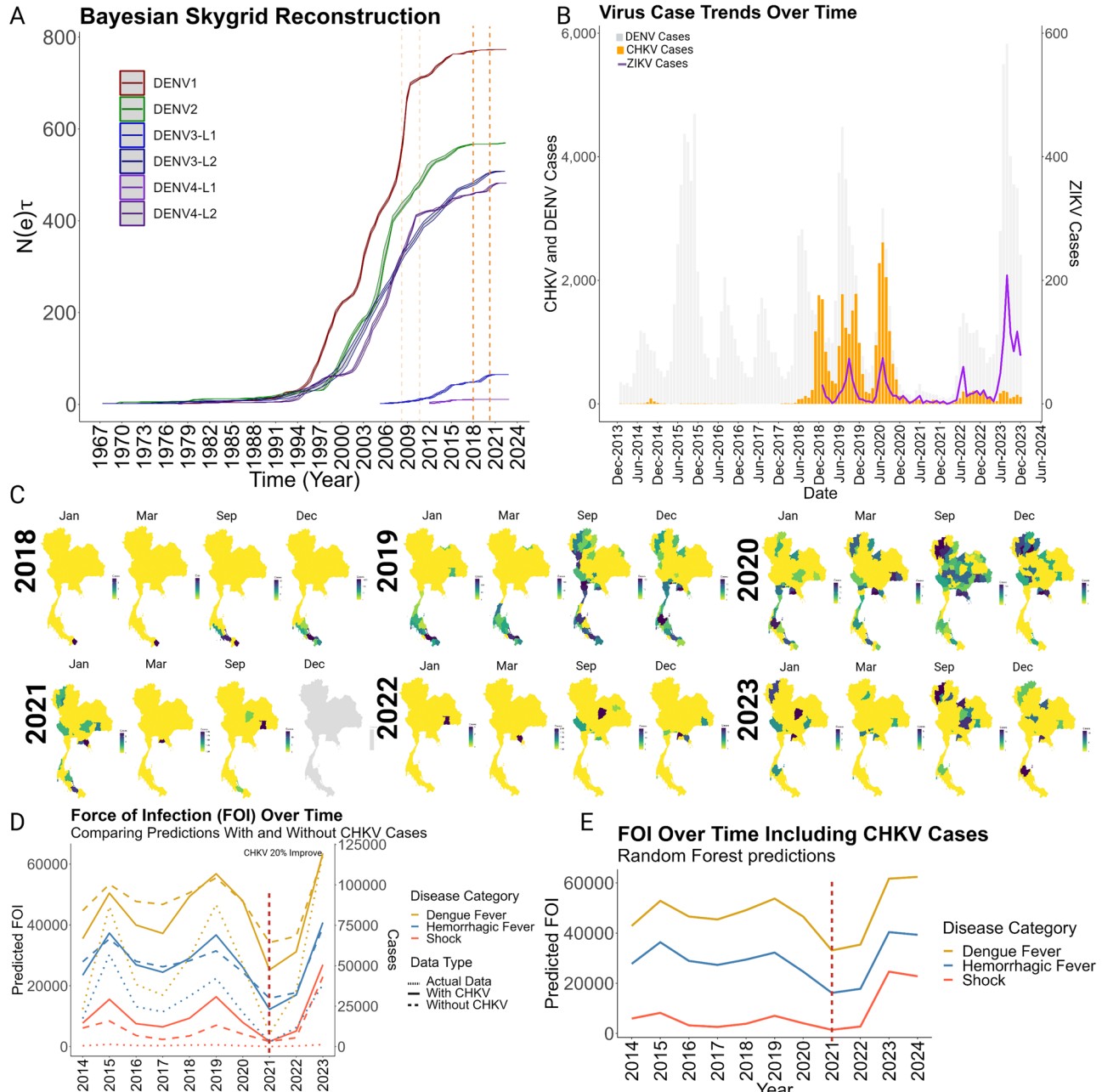

**Fig. 5 | Demographic history of DENV lineages in a comparative analysis of chikungunya (CHIKV) and zika virus (ZIKV) transmission and the impact of CHIKV on Estimated force of infection for dengue virus.** **A** Demographic reconstruction using skygrid analysis. The demographic history of the six identified DENV lineages in Thailand are shown. The major epidemic waves of Chikungunya virus (CHIKV) are marked with dashed orange lines, providing context for the temporal overlay of viral expansions. **B** Comparative analysis of chikungunya and zika virus transmission. Temporal distributions of Chikungunya (CHIKV) and Zika virus (ZIKV) cases are also shown illustrating concurrent outbreaks with DENV (in background gray). **C** Spatiotemporal distribution of CHIKV Cases. Similar

framework as for DENV were represented the transmission dynamics of CHIKV, with seasonal patterns beginning in January, peaking in March and September, and concluding in December. The comprehensive spatial analysis for CHIKV is available in Supplementary Video 3) **D** Force of infection (FOI) of DENV obtained from the random forest model compared with and without the incorporation of the CHKV cases contrasted against all cases for the different clinical categories of DENV. The drop in the cases by 2021 is denoted with a red dashed line. **E** Application of the selected model (incorporating the CHKV cases to forecast the dynamics of dengue virus infection determined by the estimated FOI for DENV over time, for the year 2024. The year 2021 is marked with a red dashed line.

To further explore the dynamic interactions between chikungunya virus (CHIKV) and dengue virus (DENV), including potential competition or partial protective effects, we conducted a detailed analysis of the geospatial distribution of CHIKV cases across Thailand. From 2018 to 2020, the geographic spread of CHIKV closely mirrored the distribution patterns of the IndexP, suggesting that regions with prior outbreaks often became the epicenters for subsequent year's epidemics (Fig. 1B and Supplementary Video 1–3). These patterns were consistent with those

observed for DENV, tracking with shifts in the *IndexP*. However, unlike the distribution trends observed with DENV, the dispersal pattern for CHIKV was notably disrupted and did not revert to its previous trajectory in the following years of 2022 and 2023 (Figs. 1C and 5B). This deviation from the expected geographic pattern could indicate factors beyond simple vector availability, such as viral interference or cross-immunity, might play critical roles in shaping the epidemiology of these viruses.

We developed a random forest model to estimate the force of infection (FOI) for dengue virus (DENV). Our variable impact analysis (Fig. 5C), supported that the infection category (dengue fever (DF), dengue hemorrhagic fever (DHF), and severe dengue fever (SDF)), yearly case data, and total case counts significantly increased node purity and therefore improved the statistical power of the model (Supplementary Fig. S3) Across all categories, the actual observed case numbers in Thailand from 2014 to 2021 matched trends estimated by FOI calculations, which use mathematical and statistical models based on epidemiologic factors and assumptions (Fig. 5D). Confident in the model's ability to predict DENV dynamics, we next assessed the potential impact of including other variables, such as concurrent chikungunya virus (CHIKV) cases. Inclusion of CHIKV case data into the model yielded a 20% improvement in its predictive ability, suggesting this factor likely influenced the observed decline in cases, notably accentuating the sharp decline in cases in 2021.The application of the random forest model to forecast DENV infection trends for 2024 predicts a moderate increase in dengue fever (DF) cases, continuing the trend observed in previous years. This rise is not only limited to DF but also extends to more severe clinical manifestations such as shock and hemorrhagic fever (Fig. 5E). Significantly, the projected values for 2024 are anticipated to surpass the incidence rates recorded in 2019. This analysis, underscores the persistent upward trajectory in both the frequency and severity of DENV cases, highlighting a critical area of concern in public health monitoring and response strategies. These results also suggest that the concurrent circulation of multiple arboviruses may drive a selective dynamic that influences the genetic composition of dengue virus strains.

## Discussion

The global incidence of dengue has significantly increased over the past two decades, representing a major public health challenge. According to the World Health Organization (WHO), there has been a ten-fold increase in reported cases worldwide, rising from 500 000 cases in 2000 to 5.2 million by 2019[38]. To some extent, this reflects awareness and improved diagnostics, but a ten-fold jump suggests something larger is afoot, as DENV cases are being reported in countries not originally thought of as endemic. The year 2019 experienced an unprecedented peak in dengue incidence, with cases reported across 126 countries[39]. In 2023, there has been a notable resurgence in dengue cases globally[40]. This resurgence is characterized by significant increases in the number, scale, and frequency of outbreaks, including expansions into previously unaffected regions[4]. Shifts in climatic conditions appear to have exacerbated the proliferation and geographic spread of dengue vectors, primarily *Aedes aegypti* and *Aedes albopictus*, to higher latitudes[41]. This expansion has facilitated the occurrence of endemic outbreaks in parts of Europe, with recent reports of local transmission in Italy[42], France[43], and Spain[44,45]. These developments highlight the critical role of environmental factors in the epidemiology of vector-borne diseases[46,47].

In our study in Thailand, the geospatial analysis revealed a significant clustering of dengue cases in central and northern regions, aligning with the trajectory of the IndexP (Fig. 1). These results are consistent with previous findings that emphasize the significant role of climatic factors and environmental conditions in influencing vector population dynamics which determine proliferation of mosquito vectors that affect the transmission rates of vector-borne diseases like dengue and other zoonotic pathogens[48]. Human population density and mobility also significantly influence the distribution of dengue[9,10]. The unusual localization of cases within central regions in the country in 2021, without widespread transmission to urban areas, raises questions about the role of acquired immunity, possibly from previous dengue infections or cross-protection conferred by other circulating arboviruses like ZIKV and CHIKV. This scenario emphasizes the complex interplay between environmental changes, human behavior, and viral ecology in shaping the patterns of dengue transmission. Considering Thailand's epidemiological conditions including co-circulation of all four DENV serotypes alongside arboviruses such as ZIKV and CHIKV, it represents a unique environment for analyzing different evolutionary processes, including inter and intra-viral competition, vector population

limitations (limited transmission capacity due to dual infections), and the impact of climate. In similar research, Brito et al.[33] explored the impact of Zika virus (ZIKV) on the transient reduction of dengue cases. The study proposed that a ZIKV outbreak could drive DENV levels lower and promote the cryptic circulation of resistant strains by providing partial, short-term immunity against dengue. Although their investigation was limited to the concurrent presence of only DENV1 and DENV2 in Brazil, their findings lay the groundwork for understanding the multifaceted dynamics between closely related arboviruses such as ZIKV and DENV. Brito et al.[33] also reported a significant surge in chikungunya virus (CHIKV) infections preceding a decline in dengue virus (DENV) cases during 2017-2018 in Brazil. Despite noting this trend, the authors dismissed the impact of CHIKV on DENV dynamics due to the taxonomic distance between the two viruses, which belong to different genera. However, subsequent research, including a study by Lima et al.[49], suggests a considerable degree of cross-reactivity between CHIKV and DENV, with reported IgM cross-reactivity percentages ranging from 36.1 to 46.7 percent, hinting at possible cross-protective immunity between these arboviruses[49]. Further evidence from multiple studies have highlighted the competitive interaction between DENV and CHIKV[50-53]. Taraphdar et al.[52] observed that CHIKV demonstrates a fitness advantage in hepatic cells, while Zaidi et al.[51] identified a replication advantage for CHIKV over DENV. In another, Tun et al.[53] reported that viremia levels in mono-infected individuals were significantly higher than those coinfected with both DENV and CHIKV, suggesting that coinfection may reduce the transmissibility of both viruses. In the present study, a random forest model was employed to examine the influence of CHIKV on DENV transmission dynamics suggesting that indeed CHIKV impacts on the force of infection of DENV (Fig. 5). Our data also revealed a synchronous decline in cases of both viruses in 2021 (Figs. 1–5), following a preceding increase, which could corroborate the earlier statements attributing this to inter-viral competition[53]. Thus, if coinfections became more prevalent, they could feasibly reduce the transmissibility of both pathogens. This phenomenon, coupled with observed declines in mosquito availability potentially linked to climatic changes (Fig. 1), could provide a comprehensive explanation for the reduction in DENV cases in Thailand during the year 2021. Our results are consistent with a recent report by Krambrich et al.[54], which demonstrated that CHIKV strains circulating during the 2018–2019 outbreak in Thailand acquired mutations in the E1/E2 spike complex, notably E1 K211E and E2 V264A. These mutations enhance vector competence, transmission efficiency, and viral pathogenicity. However, as our current findings are primarily derived from a random forest model based on temporal correlations rather than direct genetic or experimental validation, this represents a notable limitation. Future experimental studies, including in vitro validation of these genetic variants, will be essential to conclusively determine the mechanisms underlying the observed epidemiological relationship between CHIKV and DENV.

We cannot quantify nor rule out the effect that the COVID-19 lockdown measures in Thailand may have contributed to the reduction in dengue cases in 2021, by limiting human movement, reducing exposure to mosquito habitats, and potentially disrupting the dengue transmission cycle[55]. However, as we noted, these were lifted in Thailand by September 2020[56] and other countries such as Peru that have assessed the impact of COVID-19 non-pharmaceutical interventions actually reported dengue incidence increased during the pandemic[57].The observed trends in our data also revealed a correlation between the number of dengue cases and mortality rates suggesting that the fatalities were more closely linked to the volume of cases rather than the inherent severity or virulence of specific dengue virus strains. Unlike countries where only one or two serotypes predominate, Thailand is a niche for all four in relatively equal abundance. Through examining the genetic diversity and demographic characteristics of DENV in Thailand, we aimed to identify genetic features of different lineages that could confer enhanced transmission or dispersal capabilities. Our Maximum Likelihood phylogenetic and time-calibrated analyses (Fig. 2) identified predominantly homogenous clusters within the DENV serotypes, indicating stable, endemic circulation. Poltep et al.[25] reported

higher diversity levels within each serotype, however our observations were based on a comprehensive dataset from GenBank and the usage of whole genomes Our results suggest that the DENV landscape in Thailand is primarily influenced by the development and diversification of endemic lineages rather than by the continual introduction of foreign lineages (Figs. 2–4). A notable observation in the frequency and distribution of the four DENV serotypes in Thailand was the relatively balanced distribution among all four serotypes around 1993 to 2018, in contrast to other periods where one or two serotypes were distinctly dominant (Fig. 2). The shifting serotype landscape can then be potentially attributable to natural viral evolution and herd immunity dynamics. The presence of stable endemic lineages in the country could indicate that while occasional introductions of new strains has occurred[25,58,59], they have not led to significant genetic shifts among the established strains. Factors such as geographic isolation[60] and possible cross-immunity with other circulating flaviviruses may play a role in limiting the spread of novel DENV strains within the country[61]. The Bayesian Skygrid reconstruction and analysis of the effective reproduction number (Re) for DENV serotypes in Thailand provided a detailed landscape of the virus's dynamics (Fig. 5). Despite the observed variations in Neτ, the constant Re>1 across all lineages, indicative of a sustained potential for epidemic growth, highlights the enduring challenge of DENV control. The uniformity of Re also suggests that while competitive interactions among lineages might influence their genetic diversity, they do not confer a significant differential impact on the overall transmission efficiency[27].

The skygrid reconstruction and positive selection analysis revealed distinctive dynamics of diversification in endemic environments with DENV, differing notably from other endemic viral circulations where vaccinations are applied. It has been proposed that the circulation of a DENV serotype can induce herd immunity in the population, acting similarly to a vaccination and providing protection against that serotype until new ones are introduced[62]. However, with all four the dynamics observed in Thailand did not show the bottleneck effect (Fig. 5A) typically seen in other vaccinated or endemic viral populations[27,63–65]. Additionally, while positive selection analysis in viral circulation under endemic conditions or post-mass vaccination usually shows the fixation of mutations in structural proteins targeted by neutralizing antibodies[63,64], our findings point to selection of non-structural proteins, suggesting that natural selection in DENV under endemic and competitive environments favors replication and transmission over evasion of serological immune responses.

An important aspect to consider in environments where all four dengue virus serotypes co-circulate is the asymmetric competitive suppression effects described by Pepin et al.[66]. Their study provided evidence that, although DENV2 and DENV4 reached similar peak population sizes in single infections, DENV2 was more suppressed than DENV4 in mixed infections. This suggests variability in competitive abilities among different strains. Moreover, Pepin et al.[66] highlighted that competition between different dengue virus strains could significantly reduce the overall virus population size, potentially leading to decreased transmission rates. Both the Bayesian skygrid analysis and the positive selection analysis in our study revealed competitive fitness among the different serotypes. A related study by Vazeille et al.[67] demonstrated a competitive advantage of DENV4 over DENV1 in co-infections in the Aedes aegypti mosquito. A potential explanation for the major suppression of DENV2 and DENV1, compared to DENV4, is based on the directional selection observed in our pervasive analysis. Here, a higher number of sites were found that advance the replication and transmission of DENV4 compared to the other three serotypes (see Fig. 3). In scenarios where multiple DENV serotypes co-circulate, viral competition is most likely to occur in vectors where infection persists for life[68]. Co-infections can lead to unique interactions involving all aspects of the viral replication machinery. While these studies focus on competition among different serotypes and suggest benefits to the transmission of DENV4, understanding remains limited regarding inter-viral species competition, especially among different arboviruses[68].

From our positive selection analyses, several key findings emerged that reveal the evolutionary dynamics of DENV in the context of both intra- and inter-viral competition. We identified episodic selection events suggesting a synchronized evolution across three of the four serotypes at specific sites along the NS2A axis. This region, which recruits the C-prM-E-polyprotein and NS2B-NS3 in DENV1, DENV3, and DENV4, plays a crucial role in virion assembly[69]. The replacement at these selected sites likely enhances assembly efficiency, thereby potentially increasing viral transmission or fitness. Additionally, our analyses indicated positive selection in sites on the NS1 protein for all four serotypes. Given the established role of NS1 in enhancing viral acquisition by mosquitoes[70], these modifications could potentially enhance viral dissemination or confer a competitive advantage to specific serotypes at certain times within the vector. Furthermore, we observed that selected residues on the E protein in DENV1, DENV3, and DENV4 are involved in immune escape, viral-host interactions, and protein folding, among other functions. Particularly noteworthy is that all sites on the DENV3 E protein were recognized as having these functions[71](Supplementary Data 13), which could represent an adaptation of DENV3 in Thailand. The sites identified in NS5, however, were not associated with antagonizing *NF-kB* activation[72] or with changes in viral tropism[73]. This suggests that alterations in NS5 might relate more to replication efficiency. To confirm these roles, further investigations using reverse genetics and direct mutation strategies are essential.

The discrete phylogeographic analysis conducted in our study highlights the intricate roles that specific countries play within the regional and global transmission networks of dengue virus (DENV). Despite the fact that large variation of the number of sequences across countries could have impacted the results of the study, the consistency across all four serotypes indicates a minimization of the sampling bias. This analysis has been pivotal in understanding how cross-border movement significantly influences the epidemiological landscape of dengue fever[10]. Besides its favorable climate and vector availability, Thailand hosts many international visitors and migrants as a popular tourist destination and financial center in Southeast Asia. By pinpointing key nodes in this network, such as Singapore and Thailand, our findings provide critical insights that can guide public health officials in the strategic planning of interventions. Indeed, based on past experience, genotypes present in these countries will undoubtedly be exported across the globe. Interventions should be aimed not only at controlling the spread of dengue and preempting outbreaks but also at enhancing genomic surveillance. Our study underscores the complex role of regional movement in the transmission dynamics of different dengue virus serotypes (DENV1, DENV2, DENV3, and DENV4) within Thailand, highlighting its critical position in Southeast Asia's dengue ecology. The findings from Markov jump analyses reveal Thailand's significant influence on the spread of DENV1, DENV2, and DENV3, whereas DENV4's spread appears more contained to neighboring countries. This differential spread underscores the need for surveillance to detect these trends and targeted public health interventions to stem them. Awareness of mutations that may enhance the spread or severity of a lineage or alter the ability of diagnostics to detect it are perfect examples. Additionally, integrating phylogeographic and temporal data helps illuminate the complex epidemiology of dengue, emphasizing a holistic approach that combines vector control, surveillance, ecological understanding, and international cooperation. Our results also weigh the importance of considering the interactions between multiple arboviruses and the benefits of an integrated surveillance system. By using predictive modeling we described here, public health authorities can better allocate resources and strategize interventions, particularly in anticipation of potential surges in dengue cases during high-risk periods, as forecasted for 2024. This comprehensive approach is vital for reducing the burden of dengue and improving public health outcomes in the region.

## Materials and methods
### Ethics statement
This study was approved by Siriraj Institutional Review Board committee in Thailand (COA Si.391/2021). Individuals provided written informed consent. All ethical regulations relevant to human research participants were followed.

## Sample collection

Leftover plasma samples were collected during 2014-2023 from patients presenting with acute febrile illness and stored at -80°C. Individuals testing positive by dengue qPCR (n = 186) were sequenced as described below. Patients with acute febrile illness were recruited from Bangkok and Korat in Thailand between 2013 and 2023, irrespective of sex or gender. Of those whose dengue sequences were obtained and recorded, approximately 46% were female and 54% were male. All patients were of Asian descent and Thai citizenship, and their ages ranged from 15 to 72 years (median age 24). Clinical diagnoses of dengue fever or dengue hemorrhagic fever were supported by positive PCR results and confirmed via next-generation sequencing.

## Total nucleic acid extraction and cDNA synthesis

Samples were pre-treated with benzonase ( > 250 units/µl, Sigma-Aldrich, St. Louis, MO) for three (3) hours at 37°C to decrease human background nucleic acids (718 µl serum + 80 µl of 10× buffer + 2 µl Benzonase). Samples with less than the required volume were diluted with 1% PBS to obtain the required 718 µl sample volume requirement. Extraction of total nucleic acid (TNA) from 500 µl of the treated sample was performed on the Abbott Diagnostics $m$2000sp using the Sample Preparation System$_{DNA}$ (Abbott Diagnostics, Chicago). The extraction contained a positive control (five viral stocks spiked into HIV-positive plasma at 3.0 log copies/ml) and one negative control (normal human plasma). cDNA was generated from 10 µl of the extracted TNA using the SuperScript IV First-Strand Synthesis System for RT-PCR (Invitrogen, ThermoFisher Scientific) and Sequenase v2.0 DNA Polymerase (Applied Biosystems, ThermoFisher Scientific). The products were cleaned with EMnetik PCR cleanup beads (Beckman Coulter).

## Library preparation, target enrichment and DNA sequencing

NGS libraries were prepared using the Illumina Nextera XT Library Preparation Kit for 24 cycles with non-biotinylated, unique dual index barcode adapters (Integrated DNA Technologies). Libraries were purified with EMnetik PCR cleanup beads (Beckman Coulter, California). Libraries were assessed for size and quality using a TapeStation 4200 (Agilent) and quantified using a Qubit Flex Fluorometer (ThermoFisher Scientific). Next, an aliquot of each metagenomic library, with a maximum of twenty-four libraries and 3000 ng total, were pooled together for target enrichment. Pools were dried using a vacuum centrifuge, resuspended in a solution of human Cot-1 DNA and Universal Nextera Blockers (Integrated DNA Technologies), and hybridized to Comprehensive Viral Research Panel (CVRP, Twist Biosciences) probes for 16 h per manufacturer instructions. Hybridized reads were captured by affinity interaction on streptavidin beads (Twist Biosciences), amplified using a KAPA library amplification kit (Roche), and re-purified using magnetic beads (Twist Biosciences). Viral-enriched libraries were analyzed for size and concentration as before. Next generation sequencing of CVRP-enriched libraires was performed on an Illumina MiSeq instrument using a MiSeq v2 300 cycle flow cell. The resulting raw data was imported to Qiagen CLC Genomics Workbench, reads were mapped to reference sequences of all four dengue serotypes, and consensus sequences were generated.

## Genomic data collection and data curation

We retrieved all available DENV sequences from the Nextstrain repository (https://github.com/nextstrain/dengue) as of October 15th, 2024, along with their associated metadata. This dataset was augmented with 187 sequences obtained in our current study. Initially, our dataset included all sequences, regardless of whether they contained the year of isolation or the associated region/country. Therefore, the initial dataset included 14 311 sequences, covering all available sequences from Thailand and globally. All alignments are available at zenodo: https://doi.org/10.5281/zenodo.13883087.

## Sequence alignment and phylogenetic inference

In all the cases the sequences were aligned using the MAFFT software v7.453[74] with settings for a "*localpair*" option alignment. The alignments were then utilized for maximum likelihood (ML) phylogenetic inference using IQ-TREE2[75]. Initially, the *ModelFinder* algorithm was employed to identify the most appropriate nucleotide substitution model based on the Bayesian Information Criterion. Subsequently, ML phylogenetic trees were constructed. To evaluate the reliability of the branching patterns within the ML trees, we conducted 10,000 replicates each of the Shimodaira-Hasegawa Approximate Likelihood Ratio Test and Ultrafast Bootstrapping.

## Temporal and demographic Bayesian inference

Initially to determine the temporal signal of all the sequences (Global and Thailand included dataset) we focused on those sequences that had associated isolation dates. We divided the dataset of sequences corresponding to each serotype. To determine the temporal emergence and evolutionary rate of all the DENV serotypes globally and to the strains circulating in Thailand, time scaled phylogenies were generated using BEAST 1.10.5[76] an uncorrelated relaxed lognormal clock and prior distributions were set with an exponential mean of 1.0 for the clock and an exponential population size with a mean of 10.0 and an offset of 0.5 for the coalescent prior. An initial Gaussian Markov random field (GMRF) model (Gill et al., 2013) was selected for the analysis. For each DENV serotype eight independent runs were executed, each sampling from the Markov Chain Monte Carlo (MCMC) chains at intervals of $9 \times 10^8$ generations with sampling every $9 \times 10^5$ generations. In addition, the BEAGLE 3 library[77] was employed to enhance computational efficiency. Subsequent evaluation of convergence was determined by the effective sample size (ESS) parameter estimates, with values of ESS > 200, determined by using the software Tracerv1.7.

After identifying the endemic clusters within the Thai sequences, six new datasets were generated using only these sequences. Subsequently, a new temporal analysis was conducted for each lineage. To enhance resolution regarding the changes in the genetic diversity of the six endemic lineages identified in the country, we selected the SkyGrid tree prior model[78] for this analysis, while maintaining the remaining parameters as previously described.

## Estimation of the rate of infections from the different DENV lineages in Thailand using Birth-Death-skyline (BDSKY)

To estimate the effective reproduction rate (Re) of the different endemic lineages of DENV the different DENV lineages identified in Thailand we used a Birth-Death-skyline(BDSKY) model (K̈uhnertet al. 2015) included in the software package BEAST 2 v2.7.6.[79] In this model, infections transmit at a rate ($\lambda$) and transition to non-infectious status at a rate ($\delta$). Each infected individual is sampled with a probability (s) and subsequently included in the dataset. The model enables the piecewise estimation of the effective reproduction number (*Re*), $\delta$, and s over time. The non-infectious rate $\delta$ was modeled using a lognormal prior, with a mean set at 14 days and a standard deviation of 0.5, approximating the total incubation period of the dengue virus[29].

## Discrete phylogeographic reconstruction

To investigate the geographical spread of all four serotype of DENV and identify the origins of external introductions in Thailand as well as the local emergence of strains in the country, we conducted discrete trait phylogeographic inference[80], as described in Perez et al.[27]. The analyses utilized the HKY substitution model and employed a relaxed clock model with rates sourced from a log-normal distribution. Additionally, a SkyGrid coalescent model[78] was chosen as the phylogenetic tree prior. This method allows for a detailed analysis of the geographical distribution and migration patterns of the virus. For further insights, we estimated all Markov jump counts to the trees by selecting *Reconstruct complete change history on tree*, that allows to estimate both the frequency and the timing of each jump or state transition. Eight independent Markov Chain Monte Carlo (MCMC) runs were

executed, each analysis was set for $9 \times 10^8$ iterations and sampling at intervals of $8 \times 10^5$ iterations. To ensure robustness, the posterior distributions from these runs were combined after excluding the initial 10% of sampled trees from each chain. Convergence and mixing properties of the chains were evaluated using Tracer v1.7. The estimated sample size (ESS) values > 200, confirming the adequacy of the sampling and the reliability of our estimates. Subsequently, a Maximum Clade Credibility (MCC) tree was generated using TreeAnnotator v1.10. A *post-hoc* analysis was conducted to summarize the Markov Jumps estimates for transitions between discrete states, utilizing the *TreeMarkovJumpHistoryAnalyzer* tool. This tool is part of the BEAST codebase and was accessed on April 1st, 2024, from its GitHub repository. For visualization of these transitions, we employed the R package *circlize*[81].

### Epidemiological and spatial data

Data on dengue cases, categorized into three clinical severities: dengue Fever, Hemorrhagic Fever, and Shock, were obtained from the Thailand Ministry of Public Health's surveillance data portal for the years 2014 to 2023, available at http://doe.moph.go.th/surdata/disease.php?ds=66. This dataset included the number of cases and deaths per month, as well as data disaggregated by province. Similarly, monthly data for Chikungunya from 2014 to 2023 were obtained from http://doe.moph.go.th/surdata/disease.php?dcontent=old&ds=84, and Zika virus data were available from 2019 to 2023, accessed at http://doe.moph.go.th/surdata/disease.php?ds=87. Custom scripts in R were utilized to aggregate monthly dengue cases across all three categories. Additionally, spatial data for constructing vector maps and integrating the names of provinces with shapefiles were acquired using the R packages *sp*, *rnaturalearth*, *rgdal*, *raster*, and *maptools*. These tools enabled the conversion of latitude and longitude coordinates and the modification of geographical data for analysis.

### Estimation of force of infection (FOI) of dengue and the impact of CHIKV cases

In this study, we analyzed epidemiological data on dengue and Chikungunya virus cases from 2014 to 2023, categorizing dengue into three clinical severities: dengue Fever, Hemorrhagic Fever, and Shock, with records of annual total cases and deaths available at: http://doe.moph.go.th/surdata/disease.php?ds=66. Similarly, Chikungunya data included annual total cases and deaths available at: http://doe.moph.go.th/surdata/disease.php?ds=84. We employed R packages *readr* and *janitor* for standardizing variable names and formats to ensure data consistency and accuracy. We utilized a Random Forest regression model, chosen for its ability to handle non-linear relationships and multiple predictors, predicting total dengue cases as a function of time (year), disease category, and annual total Chikungunya cases, using the formula: The force of infection (FOI) for dengue, factoring in the impact of CHIKV, was estimated with the expression (1)

$$\lambda_{Dengue}(t) = \frac{\hat{D}(t|CHIKV(t))}{N_{Susceptible}} \qquad (1)$$

Where

1) $\lambda_{Dengue}(t)$ : represents the FOI of dengue at time (t)
2) $\hat{D}(t|CHIKV(t))$: represents the predicted total cases of dengue at time (t), modeled as a function of both the year, the clinical category of dengue, and the total cases of CHIKV in the same year.
3) $N_{Susceptible}$: susceptible population, assumed to be 90% based on endemism in Thailand[82]

We assumed a total population of 71 million. The model validation included k-fold cross-validation and variable importance measures.

### Estimation of mosquito-viral suitability measure

A mosquito-viral suitability measure, defined as IndexP[83], was estimated using the MVSE R-package (version 1.01)[20], incorporating local temperature, humidity, and precipitation as key climatic variables. This index assesses the reproductive potential of a single adult female mosquito within a susceptible host population[84]. We parameterized our model using satellite climate data sourced from Copernicus.eu (dataset "ERA5-Land monthly averaged data from 2014 to 2023: https://cds.climate.copernicus.eu/cdsapp#!/dataset/reanalysis-era5-land-monthly-means?tab=overview), which provided monthly observations essential for the model to fit realistic environmental conditions that influence mosquito behavior and virus transmission dynamics.

The stochastic model implemented through the MVSE package integrates empirical climatic data with theoretical assumptions about mosquito and human interactions[20]. Parameters were defined based on prior distributions reflecting our biological understanding and empirical evidence: Mosquito life expectancy (Lm), incubation period (Im), and biting rate (B) were modeled using gamma distributions with means of 10 days, 5.94 days, and 0.25 bites per day, respectively, and corresponding standard deviations of 2.55, 1.8, and 0.01 following previous estimations[84]. Human parameters such as life expectancy (Lh), incubation period (Ih), infectious period (Ph), and transmission probability from humans to mosquitoes (Bhm) were similarly modeled using gamma distributions with means of 79.1 years, 5.8 days, 5.9 days, and 0.5, and standard deviations of 2, 1, 1, and 0.01, respectively. Ecological coefficients critical for the model were estimated via a Bayesian approach, utilizing Markov Chain Monte Carlo (MCMC) simulations with 100,000 iterations and a burn-in phase to ensure robust parameter estimations[20].

### Reporting summary

Further information on research design is available in the Nature Portfolio Reporting Summary linked to this article.

## Code availability

The code to run the BEAST analyses is available in the hmc_develop branch of the BEAST codebase on GitHub (https://github.com/beast-dev/beast-mcmc). The tools TreeMarkovJumpHistoryAnalyzer and TreeStateTimeSummarizer are available from the master branch in the same codebase. The MVSE R Package is available from its home source https://webpages.ciencias.ulisboa.pt/~jmlourenco/mvse.html.

## Data availability

All the data used in the study is publicly available and has been declared in the Material and Methods section or in the Supplementary Data. All sequences and their associated metadata used in this study are available on Zenodo[85]. In addition, accession numbers for the sequences generated in this study are available in GenBank PV344255-PV344417

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

## Acknowledgements

We would like to acknowledge the Thailand government, specifically the Bureau of Epidemiology | Department of Disease Control | Ministry of Public Heath, for the epidemiologic data provided in public reports that were leveraged in this study.

## Author contributions

Conceptualization, L.J.P., P.P, Y.S, M.G.B., and G.A.C.; methodology, L.J.P., J.Y., S.W., M.A.R., and T.V.M.; software, L.J.P., J.Y., S.W. and T.V.M.; formal analysis, L.J.P., J.Y., C.C., S.W. and J M.G.B.; investigation, L.J.P., P.P, Y.S, M.G.B. resources, G.A.C.; data curation, C.C., J.Y., S.W., P.P, Y.S, and T.V.M.; writing—original draft preparation, L.J.P., Y.S, M.G.B., writing-review and editing, L.J.P., M.A.R., M.G.B., Y.S, Y.S, G.A.C.; visualization, L.J.P., and M.G.B.; supervision, M.G.B., Y.S, G.A.C, project administration, G.A.C; funding acquisition, G.A.C. All authors have read and agreed to the published version of the manuscript.

## Competing interests

Funding for this project was provided by Abbott Laboratories. The funder provided support in the form of salaries for authors L.J.P., J.Y. S.W., G.A.C.,

T.M. M.R. and M.G.B., and but did not have any additional role in the study design, data collection and analysis, decision to publish, or preparation of the manuscript.
