## [Transparent Peer Review file · Communications Biology]

Climate, inter-serotype competition and arboviral interactions shape Dengue dynamics in Thailand

Corresponding Author: Dr Lester Perez

This manuscript has been previously reviewed at another journal. This document only contains information relating to versions considered at Communications Biology.

Version 0:

Reviewer comments:

Reviewer #1

(Remarks to the Author)

I appreciate the author's responses to my prior concerns. Most of my concerns have been adequately addressed. However, several minor concerns need to be addressed:

1. Through the authors' response to comment #7, I understand that the red arrows indicate the directional progression of the cases. Based on my understanding, Figure 1E is a summary derived from observations in Figure 1D. However, the arrows in Figure 1E appear to be subjectively determined. Would the authors consider statistically identifying the center of case distributions at different period and drawing directional arrows based on their temporal progression? This approach may provide a more objective and compelling representation.
2. Figure 3: It would be better to annotate the location range for each aligned sequence.
3. Figure 5C: The legend for September 2019 is incomplete.
4. The author proposes that CHIKV outbreaks may have contributed to a reduction in DENV cases, but this conclusion relies primarily on a random forest analysis based on temporal correlations and lacks genetic-level evidence. The author should consider this limitation in the discussion.
5. The legend scales in Supplementary Videos S2 and S3 are inconsistent across their respective time scales, making it difficult to distinguish cases for the same region at different time points based on color. I recommend that the authors specify the limits parameter (e.g., `limits = c(0, 1000)`) within the `scale_fill_*` functions in `ggplot2` to ensure a consistent color scale. This adjustment will maintain a uniform mapping between colors and case counts, facilitating direct visual comparison across time.

Reviewer #2

(Remarks to the Author)

Although I'm not a biomathematician, I agree on the clarification/explanations of the authors replying to the referees. This work concerning the analysis of the four DENV serotypes evolution and the CHIKV interference from 2014 to 2023 in Thailand seems to be detailed and is of epidemiological interest, also useful for public health Institutions. It may be an advance in understanding the trend of an outbreak or an epidemic event in a given country.

REVIEWERS' COMMENTS:

Reviewer #1:

Remarks to the Author:

I appreciate the author's responses to my prior concerns. Most of my concerns have been adequately addressed.

We sincerely appreciate the Reviewer's valuable feedback and the time invested in reviewing our manuscript. The suggestions provided have undoubtedly enhanced the quality of our work.

However, several minor concerns need to be addressed:

1. Through the authors' response to comment #7, I understand that the red arrows indicate the directional progression of the cases. Based on my understanding, Figure 1E is a summary derived from observations in Figure 1D. However, the arrows in Figure 1E appear to be subjectively determined. Would the authors consider statistically identifying the center of case distributions at different period and drawing directional arrows based on their temporal progression? This approach may provide a more objective and compelling representation.

We appreciate the reviewers' insightful comment. Indeed, Figure 1E serves as a summary representation derived directly from the trajectory analyses of case progression presented in Figure 1D. To objectively analyze these trajectories, we utilized spatial statistical methods, specifically kernel density estimation (KDE) and centroid analysis, to statistically determine the central positions of case distributions across successive time periods. Directional arrows in Figure 1E were then systematically drawn based on the temporal progression of these statistically derived centroids, ensuring objectivity and reproducibility in our representation. To further enhance clarity and address the concern raised, we have incorporated detailed explanations of these statistical methods within the figure caption of Figure 1E to explicitly clarify that trajectory directions were not subjectively determined.

“The directional progression of cases were derived from statistically determined central positions (centroids) of case distributions using kernel density estimation (KDE) across successive time periods.”

2. Figure 3: It would be better to annotate the location range for each aligned sequence.

We appreciate the reviewer's suggestion; however, this request is somewhat unclear. If the reviewer is referring to annotating each individual sequence directly on their respective branches in the phylogenetic tree, we prefer to maintain the current presentation. Annotating each sequence location directly on the figure would unnecessarily clutter and overload it, potentially obscuring the key information we aim to highlight. We believe the current annotations clearly emphasize the essential aspects: the temporal progression of sequences, the identification of clades experiencing

episodic or pervasive selection, the nature of amino acid substitutions, and the protein context where these evolutionary events occurred. Including additional sequence-specific location annotations would not significantly enhance the interpretation or clarity of the results presented in Figure 3.

3. Figure 5C: The legend for September 2019 is incomplete.

We appreciate this reviewer's observation we have amended the figure

4. The author proposes that CHIKV outbreaks may have contributed to a reduction in DENV cases, but this conclusion relies primarily on a random forest analysis based on temporal correlations and lacks genetic-level evidence. The author should consider this limitation in the discussion.

We agree with the reviewer's observation and acknowledge this important limitation. However recent genetic evidence from strains of CHIKV circulating in Thailand during the 2018-2019 outbreak has identified mutations linked to increased vector competence and viral pathogenicity which are in line with our findings. To address the reviewer's comment, we have included the following clarification in the Discussion section (lines 426-435): "Our results are consistent with a recent report by Krambrich et al. (2024), which demonstrated that CHIKV strains circulating during the 2018–2019 outbreak in Thailand acquired mutations in the E1/E2 spike complex, notably E1 K211E and E2 V264A. These mutations enhance vector competence, transmission efficiency, and viral pathogenicity. However, as our current findings are primarily derived from a random forest model based on temporal correlations rather than direct genetic or experimental validation, this represents a notable limitation. Future experimental studies, including in vitro validation of these genetic variants, will be essential to conclusively determine the mechanisms underlying the observed epidemiological relationship between CHIKV and DENV."

5. The legend scales in Supplementary Videos S2 and S3 are inconsistent across their respective time scales, making it difficult to distinguish cases for the same region at different time points based on color. I recommend that the authors specify the limits parameter (e.g., $\text{limits} = c(0, 1000)$) within the `scale_fill_*` functions in `ggplot2` to ensure a consistent color scale. This adjustment will maintain a uniform mapping between colors and case counts, facilitating direct visual comparison across time.

We acknowledge the reviewer's suggestion, and indeed, we initially considered using a fixed scale across all time points. However, when applying a uniform scale, particularly for months where case numbers surpassed 6,000, it would result in masking important visual distinctions among lower case counts (e.g., between 1 and 1 000 cases). This inadvertently reduced the clarity of the visualization and achieved the opposite of our intended goal.

It is crucial to emphasize that the purpose of the maps in Supplementary Videos S2 and S3 is primarily to illustrate the geographic trajectory and spatial distribution of cases over time, rather than precise monthly quantification of cases. Maintaining individualized scales for each temporal snapshot helped to preserve clarity in visual interpretation. So after careful consideration, we have

decided to retain the original scaling strategy to best reflect the temporal and spatial dynamics to readers.

Reviewer #2:

Remarks to the Author:

Although I'm not a biomathematician, I agree on the clarification/explanations of the authors replying to the referees. This work concerning the analysis of the four DENV serotypes evolution and the CHIKV interference from 2014 to 2023 in Thailand seems to be detailed and is of epidemiological interest, also useful for public health Institutions. It may be an advance in understanding the trend of an outbreak or an epidemic event in a given country.

We sincerely thank Reviewer #2 for his/her positive assessment of our manuscript. We greatly appreciate the reviewer's recognition of the epidemiological value and public health relevance of our work, particularly regarding the evolutionary analysis of DENV serotypes and potential interference by CHIKV outbreaks in Thailand from 2014 to 2023. We also appreciate the reviewer acknowledging the clarity of our explanations in response to referee comments. Such feedback strengthens our confidence in the significance and clarity of our findings.